# Investigation of the background
# in coherent $J/\psi$ production at the EIC

**Wan Chang**[1,2,⋆]

**1** Key Laboratory of Quark and Lepton Physics (MOE) and Institute of Particle Physics,
Central China Normal University, Wuhan 430079,China
**2** Department of Physics, Brookhaven National Laboratory, Upton, NY 11973, U.S.A.

⋆ changwan@mails.ccnu.edu.cn

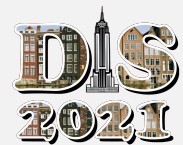

*Proceedings for the XXVIII International Workshop
on Deep-Inelastic Scattering and Related Subjects,
Stony Brook University, New York, USA, 12-16 April 2021*

## Abstract

**Understanding fundamental properties of nucleons and nuclei are among the most important scientific goals of the next-generation machine, the Electron-Ion Collider (EIC). With the unprecedented versatility provided by the EIC, it will provide answers to many standing puzzles and open questions in modern nuclear physics. One of the golden measurements at the EIC is coherent vector meson production in electron-nucleus (eA) scattering in order to obtain the spatial gluon density distribution in heavy nuclei. This requires the experiment to overcome an overwhelmingly large background arising from the incoherent diffractive production, where the nucleus mostly breaks up into fragments of particles in the far-forward direction close to the hadron beam rapidity. In this report, we systematically study the rejection of incoherent $J/\psi$ production by vetoing products from the nuclear breakup - protons, neutrons, and photons, which is modeled with the BeAGLE event generator and the most up-to-date EIC Far-forward Interaction Region design.**

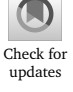

## 1 Introduction

One of the golden measurements proposed at the Electron Ion Collider (EIC) [1] is the detection of coherent and in-coherent vector-meson (VM) production from heavy nuclei [2]. This measurement has three important physics implications. Coherent production is: i) a direct measurement of the parton spatial distribution inside of a nucleus; ii) sensitive to non-linear dynamics in Quantum Chromodynamics QCD [2–5] when comparing the production of different VMs in different kinematic regions. The in-coherent VM production provides according to

the Good-Walker picture [6], the incoherent cross-section is a direct measure of the lumpiness of gluons in the ion.

The most promising channel to map the spatial gluon distribution in nuclei is to measure coherent $J/\psi$ production off a heavy nucleus, e.g., lead (Pb) in a process as, $e+Pb \rightarrow e^{'}+J/\psi+Pb^{'}$, where the scattered $Pb$ nucleus is required to stay intact. A challenging issue is that for high values of $|t|$, the momentum transfer between the incoming and outgoing nucleon (nucleus), the exclusive incoherent vector meson production, $e+Pb \rightarrow e^{'}+J/\psi+X$, overtakes the coherent production. From a recent quantitative study in the EIC Yellow Report [7], resolving the three diffractive minima of the coherent $|t|$-distribution is critical to achieve the goal of this measurement to obtain with reasonable precision the gluon density distribution. In order to observe the three minima from low to high $|t|$ at the EIC, the required rejection power is roughly 90%, 99%, and $> 99.8\%$, respectively. The UPC data at RHIC and LHC, as of now, cannot achieve the goal of measuring the gluon density distribution in a heavy nucleus, while the planned EIC-experiments with their unique detector capabilities along the beam-line might have the best opportunity in fulfilling this experimental quest in the future. This work is based on publication Ref. [8].

## 2 BeAGLE

BeAGLE is a general-purpose electron-nucleus event generator for high energy $eA$ collisions. It has been extensively used to understand the $eA$ physics and the EIC detector/interaction region design [7]. The core of the BeAGLE model is based on the PYTHIA-6 event generator [9] for simulating the parton level interactions in electron-nucleon collisions. The nuclear geometry is modeled within a Glauber-type formalism. Final-state interactions between produced particles and spectator nucleons are provided by the program of DPMJET [10]. Finally, the FLUKA model [11,12] is implemented to describe the breakup of the excited nucleus. For details, see Refs. [13,14].

The BeAGLE simulation used in this paper is based on a sample of $e+Pb \rightarrow e^{'}+J/\psi+X$ with 18 GeV electrons scattering off 110 GeV per nucleon Pb nuclei. 1.3 million events of incoherent $J/\psi$ production have been simulated.

## 3 Far-Forward Detectors

The current EIC IR and far-forward region design are based on the EIC Conceptual Design Report (CDR) [1]. The detectors are advanced concepts for measuring forward-going particles that are outside the main detector acceptance ($\theta > 35$ mrad), and are based on the EIC reference detector, detailed in the EIC Yellow Report [7]. The B0 silicon detector, off-momentum detector (OMD), Roman Pots (RP) and Zero-Degree Calorimeter (ZDC) are the four different far-forward detectors involved in this study. A pre-shower detector is also included in the set of detectors to be installed on B0 magnet bore to detect the photon with a scattering angle greater than 5 mrad and less than 22 mrad. To establish baseline particle acceptances and detector resolutions for the present study EICRoot [15] and Geant [16,17] are used in the simulations.

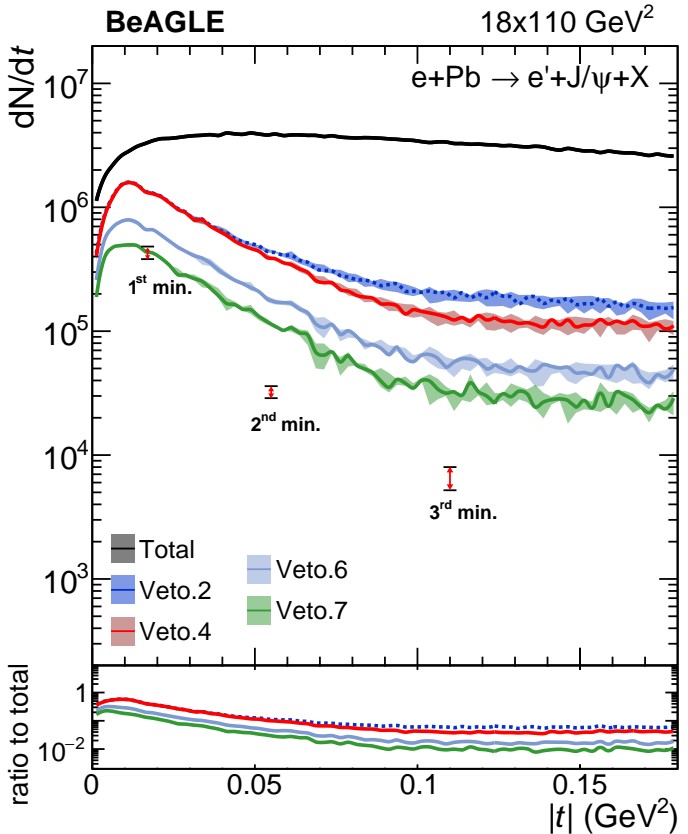

Figure 1: Distribution of the momentum transfer $|t|$ for incoherent $J/\psi$ production in $ePb$ collisions with 18 GeV on 110 GeV at the EIC. Different lines indicate results after different vetoing requirements.

## 4 Result

In the BeAGLE model, the incoherent $J/\psi$ is produced together with either protons, neutrons, photons, or any combination of them depending on the excitation energy. For a successful veto of incoherent diffractive events, detection of one particle is sufficient. Therefore, the background events remaining at the end will have none of the veto requirements fulfilled. In order to detail the vetoing procedure step-by-step, we break them down on different particles in different detectors, listed as follows:

- Veto.1: no activities ( $|\eta| < 4.0$ and $p_T > 100$ MeV/$c$) other than $e^-$ and $J/\psi$ in the main detector;

- Veto.2: Veto.1 and no neutron in ZDC;

- Veto.3: Veto.2 and no proton in RP;

- Veto.4: Veto.3 and no proton in OMD;

- Veto.5: Veto.4 and no proton in B0;

- Veto.6: Veto.5 and no photon in B0;

- Veto.7: Veto.6 and no photon with $E > 50$ MeV in ZDC.

In Fig. 1, the incoherent $J/\psi$ production $dN/d|t|$ as a function of momentum transfer $|t|$ is shown based on the BeAGLE event generator. The total number of events before any vetoing is shown as the black solid line, the other colored lines indicate the results after different vetoing requirements are applied. The results for cuts that have negligible impact on the vetoing are

not shown in the figure. The uncertainty bands are based on different results obtained varying the $\tau_0$ ($\tau_0$ is the free formation length parameter, for details, see Refs. [18, 19]), from 6 fm to 14 fm with the 10 fm used as the central value. The rejection power for different $\tau_0$ values is found to be similar, where the fraction of the total survived events after veto.7 is 1.98% for $\tau_0 = 6$ fm, and 2.14% for $\tau_0 = 14$ fm, respectively. Therefore the quoted uncertainty is less than 0.1%. No detailed studies on the vetoing power of the EIC central detector have been performed as the detector layouts are still not fully finalized. Nevertheless, vetoing particles with $|\eta| < 4.0$ and $p_T > 100$ MeV/$c$ other than the scattered electron $e^-$ and $J/\psi$ has a very small impact , because no other events categories, i.e., DIS, were included in this study. There are ~94% of these incoherent events have at least one neutron produced, and ZDC has a good acceptance for neutrons. Therefore, only less than 10% of events survived after veto.2. Because of the rigidity change, the RP and B0 made an insignificant contribution for proton measurements, while most of the protons within small scattering angle are detected by OMD. Figure 1 shows that after the vetoing on neutrons, protons, and photons, the residual contribution is about 1–10% of the total events, depending on the value of $|t|$.

Furthermore, the relative magnitude and position of the three coherent diffractive minima based on the Sar*tre* model [4] are shown by the red arrows. The difference between the upper and lower bar indicates the difference assuming a saturation and a non-saturation model [4]. The current result shown in Fig. 1 is found to be just enough to reach the first minimum. Based on the Yellow Report [7] study, not suppressing the background to the level of these minima, the Fourier transformation to obtain the gluon density distributions would be significantly smeared. So far with the current forward interaction region design and the BeAGLE model, there is at least a factor of 4 or more suppression needed to reach the second and third minimum.

## 5 Conclusion

We present an investigation of the background in coherent diffractive $J/\psi$ production using the BeAGLE event generator for 18 GeV electrons scattering off 110 GeV lead nuclei at the Electron-Ion Collider (EIC). After simulating these events using the most up-to-date EIC forward region and detectors with an beryllium beam pipe, the total vetoing fraction of these events is found to be 98%. This rejection power is found to be just enough to suppress the background events to the same level as the signal events at the first minimum position of the predicted diffractive coherent $t$ distribution, while more suppression is needed to reach the level of the second and third minimum. Although an active investigation on other possible instrumental improvements is on-going within the EIC community, the quantitative study reported in this report shows for the first time a realistic assessment of realizing this experimental measurement. The method and experimental setup employed in this work will serve as a baseline for future design iterations on the EIC forward IR design and its detectors.

## Acknowledgements

I would like to thank authors of Phys.Rev.D 104(11), 114030 (2021) for helpful discussion. I also thank T. Ullrich for discussion on the Sar*tre* model. The work of W. Chang is supported by the U.S. Department of Energy under Contract No. de-sc0012704 and the National Natural Science Foundation of China with Grant No. 11875143.

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
