# Peer review of "Investigation of the background in coherent $J/\psi$ production at the EIC"

_SciPost Physics Proceedings, doi:SciPost Phys. Proc. 8, 053 (2022)_

## Round 1 · Referee Report · Anonymous (Referee 1) · 2022-2-28

Report

The paper, Chang et al., "Investigation of the background in coherent J/ψ production at the EIC" a study of rejecting incoherent diffractive production of J/ψ by vetoing products of nuclear breakup is presented. The paper is well written, does a good job describing the motivation for the study and the event generator used in the study, and also clearly lays out the detector geometry considered for the future Electron-Ion Collider. The results in the manuscript are clear and mostly well explained.

This manuscript fulfills the requirements to be published in this journal.

There is one minor comment, in the abstract there is a line " In
this talk..." which I think should be changed to reflect that this is a written report.

I also have a comment regarding the proton vetos discussed in the Results section. In Fig. 1 the results of the study are shown with several of the different vetos applied, which are discussed in the text. However, the text mentions that vetos which have a small impact on the results are not shown. Those vetos which have a small impact are numbered 1, 3, and 5. The reasoning as to why veto 1 is not included is satisfactorily discussed. However vetos 3 and 5 are not. It might be worthwhile to add a single sentence or so describing why those vetos do not significantly impact the results. If I have missed a very obvious reason as to why this is the case, please let me know.
  • validity: -
  • significance: -
  • originality: -
  • clarity: -
  • formatting: -
  • grammar: -

Author:  Wan Chang  on 2022-03-03  [id 2263]

(in reply to Report 1 on 2022-02-28)

Dear Editors,

Thank you for forwarding to us the referee’s report on our submitted manuscript.

I thank the referee for the careful reading and the positive assessment of the manuscript. I have addressed all the comments and suggestions in the revised version and the reply submitted here.

  1. There is one minor comment, in the abstract there is a line " In this talk..." which I think should be changed to reflect that this is a written report.

Response: Thanks for spotting this point and it is now corrected.

  1. I also have a comment regarding the proton vetos discussed in the Results section. In Fig. 1 the results of the study are shown with several of the different vetos applied, which are discussed in the text. However, the text mentions that vetos which have a small impact on the results are not shown. Those vetos which have a small impact are numbered 1, 3, and 5. The reasoning as to why veto 1 is not included is satisfactorily discussed. However vetos 3 and 5 are not. It might be worthwhile to add a single sentence or so describing why those vetos do not significantly impact the results. If I have missed a very obvious reason as to why this is the case, please let me know.

Response: We would like to thank the referee for bringing up this good question. In the BeAGLE model, the incoherent J/\psi is produced together with one or more ions and, protons, neutrons, photons, or any combination of them depending on the excitation energy. ~94% of these incoherent events have at least one neutron produced, and ZDC has a good acceptance for neutrons, therefore, only less than 10% of events survived after veto.2. Because of the rigidity change, the RP has an insignificant contribution for proton measurements, while most of the protons within small scattering angle are detected by OMD. Therefore, veto.3 and veto.5 have a small impact on vetoing incoherent events. We have rephrased the sentence at the end of page.3 as follows,

Figure 1 shows, that the vetoing on protons, neutrons, and photons are all important and contribute to a significant reduction of the background. After veto.7, the residual contribution is about 1--10% of the total events, depending on the value of |t|. -> There are ~94\% of these incoherent events that have at least one neutron produced, and ZDC has a good acceptance for neutrons. Therefore, only less than 10% of events survived after veto.2. Because of the rigidity change, the RP and B0 made an insignificant contribution for proton measurements, while most of the protons within small scattering angle are detected by OMD. Figure 1 shows that after the vetoing on neutrons, protons, and photons, the residual contribution is about 1--10% of the total events, depending on the value of |t|.

---

## Round 2 · List of Changes

I have rephrased the sentence at the end of page.3 as follows,
Figure 1 shows, that the vetoing on protons, neutrons, and photons are all important and contribute to a significant reduction of the background. After veto.7, the residual contribution is about 1--10% of the total events, depending on the value of |t|.
->
There are ~94\% of these incoherent events that have at least one neutron produced, and ZDC has a good acceptance for neutrons. Therefore, only less than 10% of events survived after veto.2. Because of the rigidity change, the RP and B0 made an insignificant contribution for proton measurements, while most of the protons within small scattering angle are detected by OMD. Figure 1 shows that after the vetoing on neutrons, protons, and photons, the residual contribution is about 1--10% of the total events, depending on the value of |t|.

---

## Editorial Decision

published